# Factors Associated with Latent Tuberculosis Infection among the Hospital Employees in a Tertiary Hospital of Northeastern Thailand

**DOI:** 10.3390/ijerph17186876

**Published:** 2020-09-21

**Authors:** Patimaporn Chanpho, Naesinee Chaiear, Supot Kamsa-ard

**Affiliations:** 1Department of Epidemiology and Biostatistics, Faculty of Public Health, Khon Kaen University, Khon Kaen 40002, Thailand; patich@kku.ac.th (P.C.); supot@kku.ac.th (S.K.); 2Occupational Health and Safety Office, Faculty of Medicine, Khon Kaen University, Khon Kaen 40002, Thailand; 3Division of Occupational Medicine, Department of Community Medicine, Faculty of Medicine, Khon Kaen University, Khon Kaen 40002, Thailand

**Keywords:** latent tuberculosis infection, tuberculosis, health care worker, hospital employee

## Abstract

Latent tuberculosis infection (LTBI) can develop into tuberculosis (TB). The WHO requires the discovery and management of LTBI among high-risk groups. Health care workers (HCWs) constitute a high-risk group. Factors associated with LTBI among HCWs in Thailand need further study. The current study aimed to explore the factors related to LTBI among Thai HCWs. A hospital-based, matched case-control study was conducted. All cases and controls were HCWs at a tertiary hospital in northeastern Thailand. Between 2017 and 2019, a total of 85 cases of interferon-γ release assays (IGRAs)-proven LTBI, and 170 control subjects were selected from a hospital (two controls per case). The two recruited controls were individually matched with LTBI cases by sex and age (±5 years). Secondary data were obtained from the occupational health and safety office. Case HCWs had a higher proportion of significant factors than control HCWs (i.e., working closely with pulmonary TB—94.1% vs. 88.8%, and working in the area of aerosol-generating procedures (AGPs) 81.2% vs. 69.4%). The bivariate conditional logistic regression showed that the occurrence of LTBI in HCWs was statistically significant (*p*-value < 0.05), particularly with respect to: workplaces of AGPs (crude OR = 1.90, 95% CI: 1.01–3.58, *p* = 0.041); among HCWs performing AGPs (crude OR = 2.04, 95% CI: 1.20, 3.48, *p* = 0.007); and, absent Bacille Calmette-Guérin (BCG) scar (crude OR = 2.59, 95% CI: 1.50–4.47, *p* = 0.001). Based on the multivariable conditional logistics analysis, HCWs who performed AGPs while contacting TB cases had a statistically significant association with LTBI (adjusted OR = 1.82, 95% CI: 1.04–3.20, *p* = 0.035). HCWs who reported the absence of a BCG scar had a statistically significant association with LTBI (adjusted OR = 2.49, 95% CI: 1.65–5.36, *p* = 0.001), whereas other factors including close contact with TB (adjusted OR = 2.44, 95% CI: 0.74, 8.09, *p* = 0.123) were not significantly associated with LTBI. In conclusion, HCWs who performed AGPs and were absent a BCG scar had a significant association with LTBI, while other factors played a less critical role.

## 1. Introduction

Tuberculosis is a contagious disease caused by the bacteria *Mycobacterium tuberculosis*. The infection is transmitted person-to-person through aerosols that access the body via the respiratory system [1,2,3]. Latent tuberculosis infection (LTBI) is a state of persistent immune response to stimulation by *Mycobacterium tuberculosis* antigens without evidence of clinically-manifested active TB. A person has LTBI if they are infected with the TB mycobacteria but do not have signs of active TB disease. Although individuals with LTBI do not have active TB disease, they may yet develop the disease [4]. Studies suggest that 5 to 15% of persons with LTBI will develop active tuberculosis at some point in their lifetime, and a higher percentage if the persons are immunocompromised (i.e., 33.4% (17.4–44.2) in patients undergoing hemodialysis) [5]. WHO requires the management of LTBI in order to end the global tuberculosis (TB) epidemic (*WHO End TB Strategy*). HCWs are a high-risk group for increased risk of infection with TB and hence of developing tuberculosis (TB) diseases [1,6,7]. According to the *Global Tuberculosis Report 2019*, around 10 million people have incidents of tuberculosis, and about 1.6 million have died of tuberculosis. One in three of the world’s population has been exposed to TB, suggesting an LTBI incidence of 1.7 billion people. Thailand is on the high burden country list.

The prevalence of TB among HCWs ranges between two and six times higher than that of the general population [1,8,9,10]. In Thailand, the prevalence of TB in HCWs is 2.67 times higher than the general population [11]. In addition, 53.1% of health personnel have a latent TB infection, according to tuberculin skin testing (TST) [12,13]. Joshi et al. 2006, who studied LTBI in low and middle-income countries (LMICs), found a prevalence of 54% [9]. When considering countries with a high burden of TB, active TB among HCWs is often due to workplace exposure [14,15,16,17,18]. According to some epidemiological studies, an increased risk of infection is related to activities that involve close contact with TB patients and work in which aerosol-generating procedures (AGPs) are used. Workplaces with AGP, work duration, room size, and priority of contact with TB were also reported to be related to LTBI [14,19,20,21,22,23,24,25,26,27,28,29]. Although medical activities are known to be a risk and ineffective prevention, poor immunity (i.e., lack of an adequate Bacille Calmette-Guérin (BCG) vaccine) is considered a greater risk for having LTBI [20]. In Thailand, the BCG vaccine is a required immunization, and it is given immediately after birth [30]. The latest statistics on the coverage of BCG vaccination reported in December 2018 was 99.8% [31]. The national guideline from the Division of Tuberculosis, Department of Disease Control, Ministry of Public Health for screening TB and LTBI in HCW states that: (1) All new HCWs must undergo chest X-ray screening, and if there is no indication of TB, then LTBI screening is recommended; (2) In case the HCW is LTBI positive, TB surveillance is recommended, and a physical examination is performed every 6 months to 1 year; (3) In case the HCW is LTBI negative, the LTBI test should be repeated within 1 to 2 years; (4) Post-exposure investigation for LTBI surveillance is suggested; and (5) Preventive treatment is recommended. A few hospitals in Thailand have implemented a post exposure TB surveillance program [32].

Detection of LTBI is necessary in order to prevent the spread of tuberculosis in the workplace. Studies showed that interferon-γ release assays (IGRAs) have similar sensitivity to the TST, but have a higher specificity. In addition, TST was known to have cross-reactivity in BCG vaccinated people, therefore IGRAs are thus suitable for health personnel who have previously received the BCG vaccine [33,34,35,36,37,38]. There has been limited information regarding LTBI among HCWs in Thailand; thus, the objective of the current study was to identify the risk factors related to LTBI among HCWs occupationally exposed to pulmonary TB.

## 2. Methods

### 2.1. Study Design

This was a hospital-based, matched case-control study.

### 2.2. Study Population and Samples

HCWs in the current study included nurses, assistant nurses, and physicians. Other minor professions included laboratory technicians. Included in the study were 255 HCWs who had undergone the IGRA (QuantiFERON^®^-TB Gold-In-Tube; QFT-GIT) between 1 October 2017, and 31 October 2019. There were 85 HCW cases of LTBI and 170 HCW control subjects. The ratio of cases to controls was 1 to 2. The two recruited controls were individually-matched with LTBI cases by sex and aged (±5 years) and a simple random sampling technique was used to recruit the controls. 

#### 2.2.1. Case Definition

HCW cases worked at a 1000-bed, tertiary hospital in northeastern Thailand. They had positive QFT-GIT results between 1 October 2017, and 31 October 2019, and no history of TB or active TB or had a QFT-GIT positive result before the commencement of the study.

#### 2.2.2. Control Definition

HCW controls also worked at the tertiary hospital in northeastern Thailand. They had a negative QFT-GIT result between 1 October 2017, and 31 October 2019. A simple random sampling technique was used to recruit the samples.

### 2.3. Study Tool and Data Collection

Forms for collecting the existing data were developed. Factors associated with LTBI were collected, including personal biodata, work practice, medical and nursing practice, duration of exposure, size of the workroom, among others (Figure 1). The results of IGRA-QFT-GIT of both cases and controls were recorded on this form. IGRA indicates the use of the QuantiFERON^®^-TB Gold In-Tube test (QFT-GIT). In the current study, the IGRA result was provided by the Department of Microbiology, Faculty of Medicine, Khon Kaen University. The duration of employment was obtained from the database for human resources. 

### 2.4. Data Analyses

A description of demographic characteristics of cases and controls was presented. The characteristics of the subjects were summarized using descriptive statistics. Means and standard deviation, and medians and their ranges (minimum and maximum) were used for continuous variables. Frequency counts and percentages were used for categorical variables. A crude analysis was performed to determine the associations of factors with TBLI without controlling for confounding variables. The crude odds ratios (crude OR) and their 95% confidence intervals (95% CI) were computed by bivariate conditional logistic regression. Multivariable conditional logistic regression was used to compute adjusted odds ratios (adjusted OR) and their 95% confidence intervals (95% CI) while controlling for the effects of confounding variables. All statistical analyses were implemented using the Stata release 10 (StataCorp LLC, College Station, TX, USA).

### 2.5. Operational Definitions

*Close contact*: In this study, we categorized close contact by HCWs with TB index cases into three categories as follows: (1) performed medical or nursing activities, which were classified as aerosol-generating procedures (AGPs); (2) duration of contact and size of the working area; and (3) being classified as moderate to high and priority of contact [39,40].

#### 2.5.1. Priority of Contact with TB

The priority assigned to a contact investigation is determined according to the characteristics of the index case: susceptibility and vulnerability of contacts, and the circumstances of the exposure(s).

In summary, the priority of a TB contact was classified as ‘high’ if it was with a patient definitively diagnosed by a positive chest X-ray (CXR) or sputum acid-fast bacilli (AFB), or a patient suspected (by positive CXR) to have had a weak immune system, or the contact was via manipulated procedure (e.g., bronchoscopy, sputum induction, or autopsy). If none of the aforementioned were relevant, the duration of contact exposure and ventilation system of the room where the contact occurred were taken into account. The definition of a high priority contact was fulfilled when: (1) the cumulative exposure of a HCW to TB patients was ≥8 h in a small room with poor ventilation; (2) the cumulative exposure of a HCW to TB patients was ≥16 h in a small room with good ventilation; (3) the cumulative exposure of a HCW to TB patients was ≥24 h in a standard classroom size; and, (4) the cumulative exposure of a HCW to TB patients was ≥24 h in an open-air setting. The definition of a medium priority was fulfilled when an HCW was exposed to TB patients if no AGPs took place. To be assigned a medium priority, only the duration of exposure and room ventilation were considered (viz., cumulative exposure ≥4 h in a small room, ≥8 h in a standard size classroom, or ≥50 h in open-air). Any contact not classified as high or medium was classified as a low priority [39,40,41].

#### 2.5.2. Aerosol Generating Procedures (AGPs)

AGPs in this study included open suctioning of airway secretions, sputum induction, cardiopulmonary resuscitation (CPR), endotracheal intubation and extubation, non-invasive positive pressure ventilation (NIPPV), bronchoscopy, manual ventilation, nebulizer administration, high-flow oxygen delivery, tracheostomy, and nasogastric tube placement.

### 2.6. Ethical Considerations

This project was reviewed and approved by the Human Research and Ethics Committee of Khon Kaen University (Reference No. HE631102) and IRB00001189.

## 3. Results

A total of 85 HCWs were recruited as TBLI cases, and 170 were available for analysis as controls. The outcomes of matching procedures, percentages of cases, and controls were comparable (Table 1).

### 3.1. Demographic Characteristics

The majority of HCWs for both cases and controls were nurses and assistant nurses. They worked in the admission wards, intensive care unit, surgery ward, special ward, emergency ward, among others. The wards in which a higher proportion of cases were found were the intensive care unit and surgical wards. The mean duration of employment was 14 years (S.D.10.4 years), whereas for the control group, it was 13 years (S.D. 10.6 years). The percentage of HCWs (cases vs. controls) who worked at a place where AGPs were performed was 81.0% vs. 81.2%, respectively. The HCWs (cases vs. controls) who worked with TB patients for ≥8 was 77.7% vs. 68.8%, respectively. When working with TB cases, both the cases and controls had a similar proportion in small rooms. Sixty-five percent (65.9%) of HCW cases and 58.8% of HCW controls worked under conditions fulfilling the definition of close contact (i.e., ≥4 h in a small room, ≥24 h in a standard classroom size room, ≥100 h in a large room). A higher proportion of HCW controls were classified as having moderate to high contact priority (28.2%) compared to HCW cases (21.2%). A higher proportion of HCW cases performed AGPs (68.8%) compared to HCW controls (40.6%). The proportion of HCW cases having close contact was higher (94.1%) than HCW controls (88.8%). A higher proportion of HCW cases had absent BCG scar (52.9%) compared to HCW controls (30%) (Table 2).

### 3.2. Crude Analysis

The results of the analyses are shown in Table 3. Bivariate conditional logistic regression was used to analyze the factors associated with the occurrence of LTBI without considering the effects of other variables. The factors that were statistically significantly related to the occurrence of LTBI included HCWs who worked in a workplace where AGPs were performed (crude OR = 1.90, 95% CI: 1.01–3.58, *p* = 0.041), HCWs who performed AGPs (crude OR = 2.04, 95% CI: 1.20–3.48, *p* = 0.007), and HCWs who had absent BCG scar (crude OR = 2.59, 95% CI: 1.50–4.47, *p* = 0.001). HCWs who were categorized as having close contact with TB were associated with LTBI (Table 3).

### 3.3. Multivariable Conditional Logistic Regression

Multivariable conditional logistic regression analyses were performed, and the results showed that only two factors had statistical significance (viz., HCWs who performed AGPs while contacting TB cases (adjusted OR= 1.82, 95% CI: 1.04–3.20, *p* = 0.035), and HCWs who reported absent BCG scar (adjusted OR= 2.49, 95% CI: 1.65–5.36, *p* = 0.001)). Being in close contact with TB cases was not found to be associated with LTBI (adjusted OR = 2.44, 95% CI: 0.74–8.09, *p* = 0.123) (Table 4).

## 4. Discussion

The current study provides new information on the risk factors related to the occurrence of LTBI in Thailand, especially because there have not been any age- and sex-matched case control studies done here. Since age and sex are confounders of LTBI, [42] the result should be valid and informative. The relationship between various factors using bivariate conditional logistic regression revealed an association between risk factors and the occurrence of LTBI without considering the effects of other variables. In the current study, the factors of statistical significance associated with LTBI in the studied HCW included: (1) workplace with AGPs (crude OR = 1.90, 95% CI: 1.01–3.58, *p* = 0.041); (2) HCWs who performed AGPs (crude OR = 2.04, 95% CI: 1.20–3.48, *p* = 0.007; and (3) HCWs absent a BCG scar (crude OR = 2.59, 95% CI: 1.50–4.47, *p* = 0.001). AGPs are, by definition, high-risk procedures during which transmission of TB from patients undergoing an AGP to HCWs can occur in droplets through the air that fall on the mouth or nose or are breathed directly into the lungs [43]. Tran et al. [44] conducted a systematic review and meta-analysis of airborne transmission and found that some procedures capable of generating aerosols were associated with increased risk of SARS transmission to HCWs or were a risk factor for transmission. In a study from India, Mathew et al. [22] found that exposure to high risk procedures had a 1.65 times greater effect on LTBI among health personnel (adjusted OR = 1.65, 95% CI: 0.91–3.00, *p* = 0.090). One study, however, revealed that exposure to AGPs did not contribute significantly to the risk of TB among HCWs, but rather the use of respiratory protection when performing high-risk procedures did. Additional evidence confirmed that HCWs who did not use respiratory protection when performing high-risk procedures were at an almost three times higher risk of TB than their counterparts [28]. The current study showed that absence of a BCG scar was associated with LTBI (Crude OR = 2.59, 95% CI: 1.50–4.47, *p* = 0.001). The BCG vaccine efficiently prevents miliary and meningeal TB in children but has low efficacy against pulmonary TB in adults [24]. The presence of BCG scarring is an important indicator and confirmation of TB vaccination. Most recipients (80–85%) will have scars after vaccination [45]. A meta-analysis revealed that BCG vaccines could protect against LTBI in children under 16 years and prevent tuberculosis from LTBI [46]. Chen et al. [20] similarly reported that the BCG vaccine could protect against LTBI in HCWs and that HCWs with a BCG scar had greater protection than those who did not (adjusted OR = 0.79, 95% CI 0.65–0.9, *p* = 0.0207). As for studies in Thailand, health personnel who did not receive the BCG vaccine had a 7.60 times greater chance of contracting LTBI than those who received the vaccine [47].

When a further multivariable analysis was performed, the result showed that associations with LTBI were only HCWs who performed AGPs (adjusted OR = 1.82, 95% CI: 1.04–3.20, *p* = 0.035) and who had no BCG scar (crude OR = 2.59, 95% CI: 1.50–4.47, *p* = 0.001). Only one study in 1982 showed evidence of a relationship between AGPs and LTBI (OR = 6.15, *p* = 0.0006) [48]. The current study confirms that AGPs are a strong risk factor and should be classified as close contact, particularly since simple proximity to TB cases was not associated with LTBI (adjusted OR = 2.44, 95%CI: 0.74–8.09, *p* = 0.123). By contrast, previous reports from China suggested that there was a link between HCWs simply working closely with TB patients and infection (OR= 1.99, 95%:1.40–2.82, *p* < 0.001) [27]. The current study also showed that the risk of LTBI was related to the absence of a BCG scar (adjusted OR = 2.49, 95% CI: 1.65–5.36, *p* = 0.001) consistent with previous studies [20,45,46,47]. Using self-reporting of the presence or absence of a BCG scar in the current study could have resulted in an underestimation of BCG vaccination. A further weakness of the current study was that not all HCWs had undergone baseline LTBI screening, so generalizations about the presence and/or absence of a LTBI need further investigation.

## 5. Conclusions

HCWs performing AGPs with TB cases (adjusted OR = 1.82, 95% CI: 1.04–3.20, *p* = 0.035) and reported having absent BCG scar (adjusted OR = 2.49, 95% CI: 1.65–5.36, *p* = 0.001) were the two factors significantly associated with LTBI. The results obtained from this study suggest that HCWs who work with pulmonary TB cases should reduce exposure to an AGP or doing an AGP under a local exhaust ventilation system. In addition, a filtering face-piece respirator with at least 95% efficiency must always be worn while working with pulmonary tuberculosis. Regarding BCG vaccine booster, this study is not able to convey an inconclusive recommendation.

## Figures and Tables

**Figure 1 ijerph-17-06876-f001:**
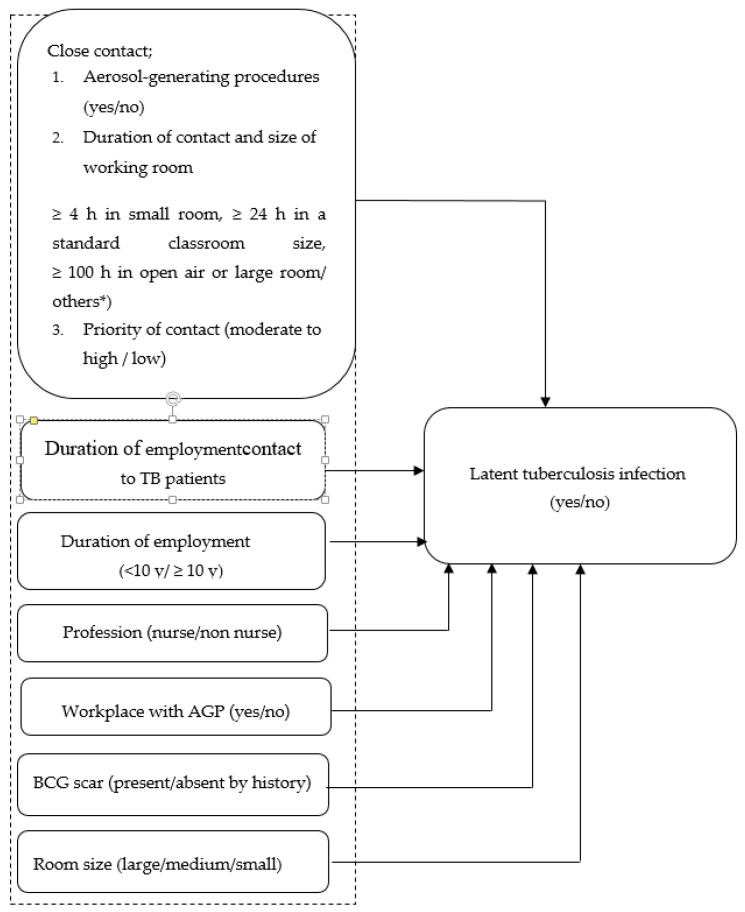
Conceptual framework of risk factors of LTBI in HCWs. * Others meant an HCW exposed to TB patients for a duration and in a working room not already defined in the assigned categories.

**Table 1 ijerph-17-06876-t001:** Outcome of case-control matching procedure.

Variables	Case		Control	
	*n* = 85	%	*n* = 170	%
Sex				
Female	79	92.9	158	92.9
Male	6	7.1	12	7.1
Age (yr)				
<30	25	29.4	50	29.4
≥30	60	70.6	120	70.6
Mean (sd)	37.5		37.4	
	(9.6)		(9.6)	
Median	36		36	
(min:max)	(25:57)		(25:57)	

**Table 2 ijerph-17-06876-t002:** Characteristics of risk factors for LTBI by cases and controls.

Characteristic	Case		Control	
	*n* = 85	%	*n* = 170	%
Duration of employment (y)	38	44.7	82	48.2
<10				
≥10	47	55.3	88	51.8
Mean (sd)	14.0 (10.4)	-	13.0 (10.6)	-

Median(min: max)	10 (1:38)	-	10 (1:37)	-

Position Nurses	53	62.4	97	57.1
Assistant nurses	26	30.6	52	30.6
others	6	7.1	21	12.4
Workplace				
Intensive care unit	27	31.8	49	28.8
Surgery ward	19	22.4	18	10.6
Medicine ward	13	15.3	21	12.4
Special ward	13	15.3	33	19.4
Emergency ward	5	5.9	28	16.5
other	8	9.4	21	12.4
Workplace of AGPs				
No	16	18.8	52	30.6
Yes	69	81.2	118	69.4
Duration of contact				
< 8 h	19	22.4	53	31.2
≥ 8 h	66	77.7	117	68.8
Close contact				
No	5	5.9	19	11.2
Yes	80	94.1	151	88.8
Priority of contact				
Low	67	78.8	122	71.8
Moderate to High	18	21.2	48	28.2
Performed AGPs				
No	35	41.2	101	59.4
Yes	50	58.8	69	40.6
Duration of contact and size of working room				
others ≥4 h in a small room, or ≥24 h in a standard classroom or ≥100 h in a large room	29 56	34.1 65.9	70 100	41.2 58.8
Room size				
Large	7	8.2	7	4.1
Medium	20	23.5	48	28.2
Small	58	68.2	115	67.7
BCG scar				
Present	40	47.1	119	70.0
Absent	45	52.9	51	30.0

**Table 3 ijerph-17-06876-t003:** Crude odds ratios for LTBI associations with various risk factors.

Characteristic	Cases (*n* = 85) %	Controls (*n* = 170) %	Crude OR	95% CI	*p*-Value
Duration					0.383
of employment (y)					
<10	44.7	48.1	1	-	
≥10	55.3	51.8	1.47	0.61–3.52	
Workplace of AGPs					0.041
No	18.8	30.6	1	-	
Yes	81.2	69.4	1.90	1.01–3.58	
Duration of contact (h)					0.148
<8	22.4	31.2	1	-	
≥8	77.6	68.8	1.53	0.85–2.79	
Close contact	-	-	-	-	0.118
No	5.9	11.2	1	-	
Yes	94.1	88.8	2.36	0.75–7.44	
Priority of contact					0.443
Low	78.8	71.8	1	-	
Moderate to High	21.2	28.2	0.67	0.36–1.27	
Performed AGPs					0.007
No	41.2	31.2	1	-	-
Yes	58.8	40.6	2.04	1.20–3.48	-
Duration of contact and size of working room					0.239
Others	34.1	41.2	1	-	
≥4 h in a small room, or ≥24 h in a standard classroom or ≥100 h in open air or a large room	65.9	58.8	1.38	0.81–2.35	
Room size		-	-	-	0.346
Large	8.2	4.1	1	-	
Medium	23.5	28.24	0.42	0.13–1.36	-
Small	68.3	67.65	0.50	0.16–1.55	-
BCG scar	-	-	-	-	< 0.001
Present	47.1	70.0	1	-	-
Absent	52.9	30.0	2.59	1.50–4.47	-

**Table 4 ijerph-17-06876-t004:** Adjusted odds ratios for LTBI associations with various risk factors.

Characteristic	Cases (*n* = 85)	Controls (*n* = 170)	Crude OR	Adjusted OR	95% CI	*p*-Value
	*n*, %	*n*, %				
Close contact						0.123
No	5, 5.9	19, 11.2	1	1	-	
Yes	80, 94.1	151, 88.8	2.36	2.44	0.74–8.09	
Duration of contact (h)						0.166
<8	19, 22.4	53, 31.2	1	1	-	
≥8	66, 77.6	117, 68.8	1.53	1.55	0.82–2.95	
Performed AGPs						0.035
No	35,41.2	101, 59.4	1	1	-	
Yes	50, 58.8	69, 40.6	2.04	1.82	1.04–3.20	
BCG scar						<0.001
Present	40, 47.1	119, 70.0	1	1	-	
Absent	45, 52.9	51, 30.0	2.59	2.49	1.42–4.38	

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
