# Peer review of "Factors Associated with Latent Tuberculosis Infection among the Hospital Employees in a Tertiary Hospital of Northeastern Thailand"

_ijerph, 2020, doi:10.3390/ijerph17186876_

Round 1

Reviewer 1 Report

Thank you for giving me a chance to review this manuscript.

This article showed HCWs with performing AGP and having BCG scar have a risk for developing LTBI in Thailand.

It is important to analyze the regional risk factor for LTBI.

I have some comments to the authors.

  1. It is unclear what behavior is defined as AGP. Please define AGP in detail.
  2. The authors counted the history of BCG as the existence of BCG scar. As the authors mentioned, 80-85 % will have scars after vaccination. Conversely, 15-20% subjects showed no scar after vaccination. Using BCG scar as a history of BCG vaccination will underestimate the proportion of real BCG vaccination. This should be mentioned in the limitation.

Author Response

Dear Reviewer I

 Thank you very much for your comments. I have attached the response to reviewer as a file attached.

Regards,

Naesinee Chaiear

Reviewer 2 Report

The article entitled “Factors associated with latent tuberculosis infection among the hospital employees in a tertiary hospital of northeastern Thailand” aims to explore the factors potentially associated with latent tuberculosis infections in healthcare workers. While the overall findings can be of interest for a vast audience, it is imperative for the authors to rewrite the manuscript paying more attention to the structure of each sentence and the wording used. I suggest using short and simple sentences that will help the reader in the comprehension of the article. Moreover, do not forget to explain or spell out abbreviations such as IGRA and BCG, which were both used throughout the manuscript without a proper explanation.

I think that the topic is interesting and useful to the scientific community. - I do not have much to say about the design of the experiment and the results shown, especially because all the statistical analyses have been performed by statisticians belonging to the epidemiology department and I am not qualified to comment on this. - Overall, I feel that this work deserves to be published, eventually.

- However, there is a general problem, almost disturbing, with the language, not only the grammar but also the syntax and the style. This is a general comment that requires a complete revision of the manuscript. I explicitly asked the authors to pay attention to the form and the language. Knowing that the the authors are not English native speakers I have suggested using a simple syntax to make the manuscript easier to read and understand.

Author Response

Dear Reviewer II

Thank you very much for your kind comments, I have attached the response to reviewer II as a file attached.

Sincerely yours,

Naesinee Chaiear

Response to Reviewer II

No

Comments

Response

Introduction

1

it is imperative for the authors to rewrite the manuscript paying more attention to the structure of each sentence and the wording used. I suggest using short and simple sentences that will help the reader in the comprehension of the article

It was rewritten as shown in

Page… 2.. line… 43-79..

2

Moreover, do not forget to explain or spell out abbreviations such as IGRA and BCG, which were both used throughout the manuscript without a proper explanation.

BCG: Page… 2.. line… 65-68..

IGRA: Page  3.. line… 85-86

                                       97-99

General comments

 However, there is a general problem, almost disturbing, with the language, not only the grammar but also the syntax and the style. This is a general comment that requires a complete revision of the manuscript. I explicitly asked the authors to pay attention to the form and the language

Thank you very much. I had the support from an English-proved reader.

Reviewer 3 Report

Title: Factors associated with latent tuberculosis infection among the hospital employees in a tertiary hospital of northeastern Thailand

Latent tuberculosis infection (LTI) may develop into tuberculosis (TB). WHO recommends the discovery of LTBI in risk groups, such as health care workers (HCW). This occupational risk in Thailand has been reported to a limited extent. This study aims to explore factors related to the occurrence of LTBI in HCW. This is a case-control study, with the collection of cases from a tertiary hospital in northeastern Thailand in the period 2017-2019. A total of 85 LTBI cases were confirmed by the ELISA IGRA test, in addition to the inclusion of 170 cases that were negative for the test. Controls were matched with cases (2:1) by gender and age (~5 years). Based on the multivariable analysis, HCW who performed aerosol-generating procedures from samples of TB cases revealed a statistically significant association with LTBI (p = 0.035), as well as those cases that did not have BCG scar (p=0.001).

The authors present a work of great relevance, especially the occupational care of health professionals who have direct contact with patients with tuberculosis. Below are some points in the manuscript that deserve to be improved.

Introduction

The authors briefly describe BCG vaccination. What is the prescription schedule for this vaccination in Thailand? Is there any data on the population's vaccination coverage?

Further, is there a national scheme for screening patients with the disease, or medical procedures for HCWs who have direct contact with the management of potentially infectious TB samples?

Page 2, lines 50-51- provide the percentage for immunocompromised patients.

Materials and methods

The methodology needs to be better represented.

The definition of case and control was made by the IGRA test, later the risk factors for the development of LTBI were established. Thus, figure 1 must be present after the sections “Case definition” and “Control definition”.

Still on Figure 1, it needs to be better detailed as the text, if not, it generates more confusion in the representation of the methodology.

The maximum criterion for the study is defined in the “close contact” section, which was divided by the authors into three categories.

  1. Performed nursing activities...ok
  2. “Duration of contact and room size”. This topic needs to be further clarified. (What is the workload of a Thai HCW? Are these hours considered weekly, monthly? Why do the authors establish ≥4 hours, ≥24 hours and 100 hours, if the tables adopted criteria of <8 hrs and ≥8 hrs? What do the authors consider as small room, medium room and large room?
  3. “Being classified as moderate to high and priorities of contact”. The subtitles “index case classification” and “priorities of contact to TB” appear to be part of this category, or are complementary. This methodology can be summarized and better clarified. For example: The high-risk professional was considered, the one exposed (handling) to the patient who had a positive chest X-ray or sputum acid fast bacilli (AFB), etc. Another question, did these health professionals use respiratory protection in the workplace? in management?

Results

Page 5, line 183-184 - are these hours monthly?

Table 2 - 6 cases and 21 controls are professionally categorized as “others”, this could be better clarified in the methodology. Are they laboratory professionals, doctors?

Table 2 - “Duration of contact and working room size” - “others - 29 cases and 70 controls” What would that others be?

Table 3 - “Duration of contact and working room size” - “others - 34.1 cases and 41.2 controls” What would that others be?

Minor points

Page 4 - line 126 and 130 - has the word employee been removed? is a correction? Because it appears scratched.

Page 9 - Lines 223-227 - “The result may be explained… into the lungs without protection” - This sentence could be reformulated.

Author Response

Dear Reviewer III

Thank you very much for your valuable comments. I have replied and clarified following your comments as a file attached.

Sincerely yours,

Naesinee Chaiear

Response to Reviewer III

No

Comments

Response

Introduction

1

The authors briefly describe BCG scar. What is the prescription schedule for this vaccination in Thailand? Is there any data on the population’s vaccination coverage?

Thailand officially launched its nation-wide immunization program (EPI) in 1977 . Currently, the Thai EPI includes vaccines that cover the following 10 antigens: tuberculosis (BCG), hepatitis B, diphtheria, tetanus (TT), pertussis, poliomyelitis (OPV), measles, mumps, rubella, and Japanese encephalitis (JE). Apart from the infant EPI vaccines, flu vaccine has been given to health care workers since 2004 and to people with certain chronic diseases since 2008.

Ref. Muangchana C, Thamapornpilas P, Karnkawinpong O. Immunization policy development in Thailand: The role of the Advisory Committee on Immunization Practice. Vaccine 2010; 28S:A104-A109.

Added: page..2   Line…67-68.

-Coverage of BCG surveyed in Dec 2018, 99.8%.

Ref: Division of Vaccine Preventable Diseases, Department of Disease Control. Report of national immunization coverage survey 2018. Nontaburi; Division of Vaccine Preventable Diseases; 2019. Available from; https://ddc.moph.go.th/uploads/files/09f76ef981ed953a6687ff5ed551af6e.pdf

Added: page 2 .line…67-68….

2

Is there a national scheme for screening patients with the disease, or medical procedures for HCWs who have direct contact with the management of potentially infectious TB samples?

There is the national guideline from Division of Tuberculosis, Department of Disease Control, Ministry of Public Health for screening TB and LTBI in HCW. Details include 1) all new HCW must have chest x ray screening and if no TB disease then LTBI screening is recommended. 2) in case of LTBI positive, TB surveillance is recommended for physical examination every 6 months to 1 year, 3) in case of LTBI negative, repeating LTBI test with in 1 or 2 years, 4) post exposure investigation for LTBI surveillance is suggested and 5) a preventive treatment is also recommended. [49]

Added: page…2…line…68-79

3

Page 2, lines 50-51 provide the percentage for immunocompromised patients.

Added: 33.4% (17.4–44.2) in patients undergoing hemodialysis

Added: page…2…line…50

Methodology

4.

The definition of cases and control was made by IGRA test later the risk factors for the development of LTBI were established. Thus, figure 1 must be present after the sections ‘case definition’ and ‘control definition’

Reorientation of the figure 1

5.

Figure 1. it needs to be better detailed as the text , if not, it generates more confusion in the representation of the methodology.

Added duration of contact  < 8 h / ≥  8 h and room size into the Figure 1

6.

The maximum criterion for the study is defined in the “close contact” section, which was divided by the authors into three categories.

-“Duration of contact and room size”. This topic needs to be further clarified. (What is the workload of a Thai HCW? Are these hours considered weekly, monthly? Why do the authors establish ≥4 hours, ≥24 hours and 100 hours, if the tables adopted criteria of <8 hrs and ≥8 hrs? What do the authors consider as small room, medium room and large room?

-          Reply: In Thailand, working hours of medical professions are more than 48 h per week.

Reply:  The establishment of  ≥4 hours, ≥24 hours and 100 hours must link with the room size while working as recommended by National Tuberculosis Controllers Association; Centers for Disease Control and Prevention. Guidelines for the investigation of contacts of persons with infectious tuberculosis. Recommendations from the National Tuberculosis Controllers Association and CDC. MMWR Recomm. Rep. 2005, 54, 1–47 and O’Malley, M.; Brown, A.G.; Colmers, J.M. Maryland TB guidelines for prevention and treatment of tuberculosis; Maryland Department of Health and Mental Hygiene: Baltimore (MD), 2007.

           For including of  8 h duration, since 8 h is generally known of a close contact without considering of other factors

-          What do the authors consider as small room, medium room (a standard classroom size) and large room?

Reply: In the questionnaire, there are the clarification of the room types;

Large room size: a kind of conference or lecture hall

Standard class room size /medium  room size: a kind of admission ward

Small room size: a kind of isolation room

-“Being classified as moderate to high and priorities of contact”. The subtitles “index case classification” and “priorities of contact to TB” appear to be part of this category, or are complementary.

1. This methodology can be summarized and better clarified. For example: The high-risk professional was considered, the one exposed (handling) to the patient who had a positive chest X-ray or sputum acid fast bacilli (AFB), etc.

2.Another question, did these health professionals use respiratory protection in the workplace? in management?

1.      Operational definitions were rewritten according to the reviewer’ suggestion.

Added: page…4..line…120-145…

2. The hospital provide N95 when taking care TB patients, however, for exposure criteria, both cases and controls did not wear a proper respirator.

Results

Page 5, line 183-184 - are these hours monthly?

These are an accumulation of hours working with TB patients.

Table 2 - 6 cases and 21 controls are professionally categorized as “others”, this could be better clarified in the methodology. Are they laboratory professionals, doctors?

 Added in page…3..line…84-85

Table 2 - “Duration of contact and working room size” - “others - 29 cases and 70 controls” What would that others be?

Explained in page 5 line 179-180

Table 3 - “Duration of contact and working room size” - “others - 34.1 cases and 41.2 controls” What would that others be?

 Explained in page 5 line 179-180

Minor points

Page 4 - line 126 and 130 - has the word employee been removed? is a correction? Because it appears scratched.

It was rewritten as page..3.. line.. 84-88…

Page 9 - Lines 223-227 - “The result may be explained… into the lungs without protection” - This sentence could be reformulated.

It was reformulated into page..9.. line.. 237-239…

Round 2

Reviewer 2 Report

I appreciate the effort put by the authors in the rewriting of the article. Now the article is well-written and the data presented clearly.

Minor comments:

  1. Please spell out IGRA also in the abstract.
  2. In line 76 you state that IGRAs have similar sensitivity to the TST, but have higher specificity. Can you please clarify in the text what else TST detect?
  3. The layout of the section 2.2 is off.

Author Response

Dear Editors,

Thank you very much for your prompt reply and support, I indeed had a geart opportunity to work with IJERPH. 

Following, the reviewer II's comments, I have replied and corrected three points and minimal change also with blue hilight.

   In case a further clarification is required please do not hesitate to get back to me.

Looking forward to hearing from you in the near future.

  Sincerely yours,

  Naesinee  Chaiear
